# Profiles of Polyphenol Intake and Type 2 Diabetes Risk in 60,586 Women Followed for 20 Years: Results from the E3N Cohort Study

**DOI:** 10.3390/nu12071934

**Published:** 2020-06-29

**Authors:** Nasser Laouali, Takiy Berrandou, Joseph A. Rothwell, Sanam Shah, Douae El Fatouhi, Francesca Romana Mancini, Marie-Christine Boutron-Ruault, Guy Fagherazzi

**Affiliations:** 1Center for Research in Epidemiology and Population Health (CESP), Institut Gustave Roussy, U1018 Inserm, 94800 Villejuif CEDEX, France; Joseph.ROTHWELL@gustaveroussy.fr (J.A.R.); sanam.shah@eleve.ehesp.fr (S.S.); Douae.EL-FATOUHI@gustaveroussy.fr (D.E.F.); Francesca.MANCINI@gustaveroussy.fr (F.R.M.); Marie-christine.BOUTRON@gustaveroussy.fr (M.-C.B.-R.); guy.fagherazzi@gmail.com (G.F.); 2Faculty of Medicine, Paris-South Paris Saclay University, 94800 Villejuif, France; 3Cardiovascular Research Center, University of Paris, UMR 970 Inserm, 75015 Paris, France; takiy.berrandou@inserm.fr; 4Digital Epidemiology Hub, Department of Population Health, Luxembourg Institute of Health (LIH), 1445 Strassen, Luxembourg

**Keywords:** diet, polyphenols, type 2 diabetes, epidemiology, E3N cohort

## Abstract

Most studies on dietary polyphenol intake and type 2 diabetes (T2D) risk have focused on total or specific subclasses of polyphenols. Since polyphenols are often consumed simultaneously, the joint effect of an intake of multiple subclasses should be explored. We aimed to identify profiles of the dietary polyphenol subclasses intake associated with T2D. A total of 60,586 women from the Etude Epidémiologique auprès de femmes de l’Education Nationale (E3N) cohort study were followed for 20 years between 1993 and 2014. T2D cases were identified and validated. The individual energy-adjusted daily intakes of 15 subclasses of polyphenols were estimated at baseline using a food frequency questionnaire and the PhenolExplorer database. We used Bayesian profile regression to perform the clustering of the covariates by identifying exposure profiles of polyphenol intakes and, simultaneously, link these to T2D risk by using multivariable Cox regression models. We validated 2740 incident T2D cases during follow-up, and identified 15 distinct clusters with different intake profiles and T2D risk. When compared to the largest cluster (*n* = 6298 women), higher risks of T2D were observed in three of those clusters, which were composed of women with low or medium intakes of anthocyanins, dihydroflavonols, catechins, flavonols, hydroxybenzoic acids, lignans, and stilbenes. One cluster (*n* = 4243), characterized by higher intakes of these polyphenol subclasses, exhibited lower T2D risk when compared to the reference cluster. These results highlight the importance of a varied diet of polyphenol-rich foods such as nuts, fruits, and vegetables to prevent T2D risk.

## 1. Introduction

Type 2 diabetes (T2D) is a lifelong disabling disease and its global prevalence continues to increase rapidly. Emerging evidence suggests that inflammation and oxidative stress play a key role in the pathogenesis of T2D [1,2]. Diet is one of the main lifestyle-related factors which can modulate the inflammatory process [3,4], and it is well known that the consumption of certain foods and nutrients is able to elicit immunomodulatory effects [5,6]. Polyphenols, bioactive compounds found in fruits and vegetables, have biological activities [7] which include modulation of the inflammatory signaling cascades and improved vascular function [7,8,9]. The role of polyphenol intake in the etiology of T2D has been widely investigated. Prior studies on dietary polyphenol intake and T2D risk reported a lower risk for higher total polyphenol intake, while mixed results were found in other studies that focused on specific subclasses of polyphenols [10]. Since polyphenols are ubiquitous in plant foods and beverages and are therefore often consumed simultaneously, clustering approaches that account for highly correlated intakes are of interest to better clarify the associations between polyphenol intake and diabetes risk. Therefore, we aimed to identify profiles of the polyphenol intake associated with T2D incidences in 60,586 French women.

## 2. Materials and Methods

### 2.1. Study Population and Follow-Up

The protocol of the Etude Epidémiologique auprès de femmes de l’Education Nationale (E3N) cohort, a French prospective cohort study, has been already described [11]. Briefly, 98,995 women were recruited from those affiliated to the French national health insurance plan for teachers and coworkers, the Mutuelle Générale de l’Education Nationale. The women were enrolled and subsequently followed by biannual self-administered questionnaires on health conditions, lifestyle, diet, treatments, mental health status, etc. Furthermore, for each cohort participant, the health insurance plan provided data that included all outpatient reimbursements for health expenditures since January 1, 2004; these data included medication brand names, doses, and dates of reimbursements. The average response rate to a follow-up questionnaire was 83%, with a total loss to follow-up since 1990 below 3%.

Eligible participants (*n* = 74,522) were those that completed the dietary questionnaire sent in 1993. Then we excluded all prevalent T2D cases (*n* = 824), women with extreme energy intakes (i.e., below the 1st and above the 99th percentiles of the energy intake over energy requirement distribution in the population) (*n* = 1491), women who did not complete any follow-up questionnaire after the dietary questionnaire (*n* = 1216) and women with missing data on polyphenol intake and covariables (*n* = 10,405). The final study population included 60,586 women. Follow-up started in 1993 (the baseline for the present study) and ended in 2013 (the latest date of T2D case validation in the E3N cohort).

### 2.2. Polyphenol Intake

Food intake was assessed using a validated 208-item semi-quantitative dietary questionnaire sent in 1993 [12]. All food intakes were converted into intakes of energy and nutrients using food composition databases from the French Information Center on Food Quality [13] and the Phenol-Explorer database [14]. Energy adjusted intakes of 15 subclasses of polyphenols were computed using the residual method as proposed by Willett et al. [15].

### 2.3. Ascertainment of Type 2 Diabetes

The detailed procedure has been described in detail elsewhere [16,17]. Subsequently, before 2004, all potential cases of type 2 diabetes were identified through follow-up questionnaires that included questions on the diagnosis of diabetes, diabetes-specific diets, diabetes drugs and hospitalizations for diabetes. All potential cases were then contacted and asked to answer a diabetes-specific questionnaire that included questions on the circumstances of diagnosis (year of diagnosis, symptoms, biological examinations, and fasting or random glucose concentration at diagnosis), diabetes therapy (prescription of diet or physical activity, list of all glucose-lowering drugs already used), and the most recent concentrations of fasting glucose and HbA_1c_. In order to be considered as validated for type 2 diabetes, an individual must have reported at least one of the following: (1) fasting plasma glucose ≥ 7.0 mmol/L or random glucose ≥ 11.1 mmol/L at diagnosis; (2) use of a glucose-lowering medication; (3) most recent values of fasting glucose concentrations ≥ 7.0 mmol/L or HbA_1c_ level ≥ 53 mmol/mol (7.0%) in the diabetes-specific questionnaire. After 2004, the identification of cases was based on the drug reimbursement insurance database. All women with at least two reimbursements for any glucose-lowering medication within a 1-year period were considered to be validated diabetes cases, with the date of diagnosis defined as the date of their first reimbursement.

### 2.4. Covariables

All potential covariates were selected a priori because of their known or suspected association with diabetes status and/or polyphenol intakes. In general, the covariables used were measured at baseline (1993) but if not available, we used the measurement at the closest wave for all women. BMI was calculated by dividing weight in kilograms by height in meters squared and was considered as continuous in all models, and in three categories for stratified analyses (<20 kg/m²/20–25 kg/m²/and ≥25 kg/m²). The level of recreational physical activity in metabolic equivalent of task (MET-h/week) was considered as a continuous variable. We considered the following categories: never, former, and current for smoking status; undergraduate or less, graduate, and postgraduate or more for education level; yes or no for personal history of hypercholesterolemia (wave 7), personal history of hypertension (wave 9), and family history of diabetes (wave 8). Alcohol, caffeine, and total energy intakes were considered as continuous.

### 2.5. Statistical Analysis

Baseline characteristics of the participants overall and according to T2D status were expressed as means ± standard deviation (SD), or numbers (percentage) for categorical variables. The correlations between the energy-adjusted intakes of polyphenol subclasses were explored by the Spearman’s rank correlation analysis.

For our main analysis, we used Bayesian profile regression [18], which is based on Dirichlet process mixture model methods [19]. This method allows for partitioning observations into clusters using two sub-models, an assignment sub-model for covariates (exposures of interest), and a disease sub-model, fitted jointly using Markov chain Monte Carlo methods. In the assignment sub-model, clustering was based on the tertile ranges of intakes of individual polyphenol subclasses. For the disease sub-model, we used Cox proportional hazards regression models adjusted for potential confounders to estimate hazard ratios (HR) and 95% CI of T2D risk. The profile regression presents the advantage of assessing collinear variables in one analysis model which cannot be done with standard regression models. Moreover, this clustering approach is advantageous because the number of clusters is not fixed in advance but determined throughout the algorithm. The profile regression was implemented with the PReMiuM package in R [20]. Further details on the Bayesian profile regression approach have been reported [20,21].

The associations between the total polyphenol, intake of each subclass and T2D risk were also estimated individually by multivariable Cox regression models. Participants were followed from their age at baseline until their age at diagnosis of T2D, at death, at the last follow-up, or at the end of the follow-up period (2014), whichever occurred first. All models were adjusted for age, physical activity, smoking status, level of education, BMI, family history of diabetes, personal history of hypertension, hypercholesterolemia, and alcohol, caffeine, and energy intakes. The selection of confounders was made a priori, based upon the known risk factors of T2D available in our dataset and associated with polyphenol intake.

In addition, we investigated the foods and beverages that contributed most to the polyphenol intake profiles and reported median (interquartile range) intakes across profiles.

Statistical analyses were performed using SAS 9.4 (SAS Institute, Inc., Cary, NC, USA) and R.

### 2.6. Ethical Approval

The study was approved by the French National Commission for Data Protection and Privacy (ClinicalTrials.gov identifier: NCT03285230). All participants gave their written informed consent.

## 3. Results

### 3.1. Baseline Characteristics

Over a mean (SD) follow-up of 18.81 (4.3) years, a total of 2740 (4.5%) validated incident T2D cases were identified. Table 1 presents the baseline characteristics of the study, overall and according to T2D status. The mean (SD) age of the population at baseline was 53 years (6.6). There was a strong correlation between the subclasses of polyphenol intake (Appendix A).

### 3.2. Clusters of Polyphenol Subclasses Intake and Type 2 Diabetes Risk

The profile regression identified 15 distinct clusters with a unique profile of polyphenol subclasses intake and T2D risk (Figure 1). Characteristics of each cluster are summarized in Appendix A.

When compared to the mean logHR of the largest cluster (cluster 5), cluster 15, composed of women with high intakes of anthocyanins, dihydroflavonols, catechins, flavonols, hydroxybenzoic acids, lignans, and stilbenes, were at lower risk of T2D (logHR = −0.004 (95%CIr −0.002,−0.007)). When compared to the mean logHR of cluster 5, clusters 1 (logHR = 0.002 (95%CIr 0.004, 0.000)), 2 (logHR = 0.001 (95%CIr 0.003, −0.001)), and 3 (logHR = 0.001 (95%CIr 0.003, −0.001)), characterized by low intakes of these polyphenol subclasses, exhibited higher T2D risk. When analyzed separately, intakes of total polyphenols and of all subclasses (except for catechins and hydroxybenzoic acids) were inversely associated with T2D risk (Appendix A).

Cluster 15 was characterized by high consumptions of fruits, vegetables, olive oil, and wine compared to the high-risk clusters (clusters 1, 2, and 3). The most important food contributors to polyphenol intakes were reported for clusters 1, 2, 3 and 15 (Appendix A).

## 4. Discussion

In this prospective cohort study of 60,586 women, we identified profiles of the polyphenol intakes associated with T2D risk. The clusters composed of women with low or medium intakes of anthocyanins, dihydroflavonols, catechins, flavonols, hydroxybenzoic acids, lignans, and stilbenes were at slightly higher risk of T2D than the reference group, and the group characterized by high intakes of these polyphenol subclasses had a lower risk of T2D, independent of major potential confounders. The cluster associated with a low risk of T2D was characterized by high consumptions of fruits, vegetables, olive oil, and wine compared to the high-risk clusters.

Polyphenol intake and its roles in public health have gained much attention in the framework of the World Health Organization’s goal to reduce and prevent chronic diseases. Previous epidemiological studies have reported inverse associations between the intake of total and specific polyphenol subclasses and T2D risk. However, no study has so far identified profiles of the polyphenol intake associated with T2D risk. Dietary polyphenols represent a large group of bioactive molecules widely found in foods of plant origin, and their health protective effects have been mainly related to their antioxidant and anti-inflammatory properties. However, it must be considered that the application of isolated polyphenols as nutraceuticals is quite limited due to their poor systemic distribution and low to medium bioavailability [22]. Traditionally, studies have focused on total polyphenol intake or intakes of specific subclasses. In this way, the associations between total polyphenol, intake of each subclass and T2D risk were estimated. For example, in our study, catechins and hydroxybenzoic acids contribute to T2D risk clusters as 85% and 96% of women in cluster 15 have high intakes of catechins and hydroxybenzoic acids, respectively, but when analyzed individually, those polyphenols were not associated with T2D risk. This may explain the null results in some studies that consider only one polyphenol subclass at a time in the analyses [10].

The majority of polyphenol-rich foods were consumed in high proportions in the low-risk cluster. These foods are important contributors to dietary patterns such as the Mediterranean diet and the dietary inflammatory index, which have been associated with a decreased risk in T2D in several studies [17,23].

Broadly, the favorable effects of polyphenols have been attributed to their anti-inflammatory effects and interactions with signaling proteins [24]. Polyphenols inhibit the function of luminal disaccharidases which reduce fasting and post-prandial hyperglycemia as well as increase glucose uptake in muscle and adipocytes to lower blood glucose [25]. In addition, polyphenols modulate the function of the liver which plays an important role in the regulation of blood glucose levels [25]. As regards the anti-inflammatory effects of polyphenols, it is mostly achieved by blocking the expression of inflammatory cytokines [26,27]. Thus, through diverse pathways, polyphenols could play an important role in T2D risk reduction.

The main strength of this study was the assessment of the cluster effects of polyphenols which is more relevant than studying individual polyphenols. Furthermore, the prospective nature of the study design, large sample size, and long follow-up time add strength to our findings. However, there are some limitations such as dietary data being only available at baseline, thus not allowing for the consideration of dietary changes during follow-up. In addition, our analyses are based on self-reported dietary consumption. In order to minimize the potential measurement error in the usual diet, we used a validated tool [12] and women with implausible diets were excluded. Finally, results from this study may not be generalizable to other populations. 

## 5. Conclusions

In this study, we identified profiles of the polyphenol intakes associated with slightly increased or decreased T2D risks in women. The cluster at low risk was characterized by high consumptions of fruits, vegetables, olive oil, and wine compared to the clusters at high risk. This result raises the importance of a varied diet of polyphenol-rich foods such as nuts, fruits, and vegetables to prevent T2D risk.

## Figures and Tables

**Figure 1 nutrients-12-01934-f001:**
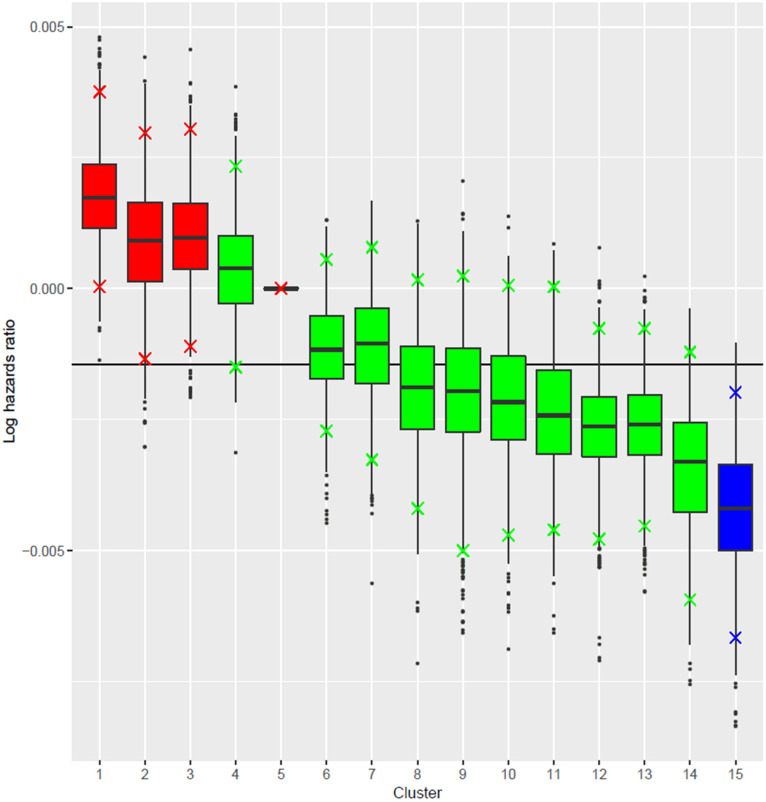
Log hazard ratios (logHRs) of clusters of polyphenol intakes profiles. The reference cluster is the largest cluster (cluster 5). The horizontal line represents the logHR mean for each analysis. The clusters whose 95% credible interval include the logHR mean are colored in green, and the rest in red for clusters at higher risk or blue for the cluster at lower risk.

**Table 1 nutrients-12-01934-t001:** Baseline characteristics of the Etude Epidémiologique auprès de femmes de l’Education Nationale (E3N) study population overall and by diabetes status.

		Type 2 Diabetes Diagnosed during the Follow-Up	
	Overall (N = 60,586)	No (N = 57,846)	Yes (N = 2740)
Age at baseline (years)	52.69 (6.62)	52.62 (6.62)	54.11 (6.62)
Educational level (%)			
Undergraduate or less	6044 (9.98)	5675 (9.81)	369 (13.47)
Graduate	31,512 (52.01)	29,971 (51.81)	1541 (56.24)
Postgraduate or more	23,030 (38.01)	22,200 (38.38)	830 (30.29)
Physical activity (MET-h/week)	49.24 (49.88)	49.26 (49.83)	48.81 (51.04)
Smoking status (%)			
Current	8295 (13.69)	7867 (13.60)	428 (15.62)
Former	20,256 (33.43)	19,391 (33.52)	865 (31.57)
Never	32,035 (52.88)	30,588 (52.88)	1447 (52.81)
BMI (kg/m^2^)	22.88 (3.23)	22.70 (3.04)	26.55 (4.58)
Hypertension (%)	30,974 (51.12)	28,937 (50.02)	2037 (74.34)
Hypercholesterolemia (%)	4273 (7.05)	3870 (6.69)	403 (14.71)
Family history of diabetes (%)	6821 (11.26)	6194 (10.71)	627 (22.88)
Alcohol consumption (g/day)	11.64 (13.85)	11.61 (13.76)	12.16 (15.53)
Coffee intake (ml/day)	200.81 (148.26)	200.89 (148.01)	199.12 (153.51)

N (%) for categorical variables and Mean (SD) for continuous variables. Abbreviation: Metabolic equivalent of task (MET).

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
