# Peer review of "Profiles of Polyphenol Intake and Type 2 Diabetes Risk in 60,586 Women Followed for 20 Years: Results from the E3N Cohort Study"

_nutrients, 2020, doi:10.3390/nu12071934_

Round 1
Reviewer 1 Report
This is a concise and well-written manuscript by Laouali et al. aiming to identify profiles of dietary polyphenol subclasses intake associated with risk of type 2 diabetes in a large French prospective cohort study.
The manuscript is of interest although some major criticisms should be addressed:
- Ascertainment of cases. Since data are self-reported, it is not clear to me how data on values of fasting glucose or HbA1c concentrations were gathered.
- If I understood well, there have been two different methods for defining incident of type 2 diabetes. Is this methodologically sound?
- I am not sure about the selection of dietary covariates. Authors controlled analyses for alcohol and coffee but they do not explain the reason. Moreover, I suggest the authors use a more global measurement of the diet quality, as for example a healthy dietary score.
- I cannot see any sub-group analysis. It would be interesting to test the robustness of main findings.
- The limitations of the study should be implemented also by clearly stating the limitation of self-reported dietary data.
Author Response
Response Letter
Dear Dr. Yoshimi Kishimoto,
Thank you for giving us the opportunity to submit a revised draft of our manuscript
entitled "Profiles of polyphenol intake and type 2 diabetes risk in 60,586 women followed for 20 years: results from the E3N cohort study" to nutrients. We appreciate the time and effort that you and the reviewers have dedicated to providing valuable feedback on this manuscript, and we are grateful to the reviewers for their insightful comments.
This revised manuscript incorporated changes to reflect the suggestions provided by the
reviewers. In this revised version, any revisions have been clearly highlighted using the
“Track Changes” function in Microsoft Word.
Here are our point-by-point responses to the reviewers’ comments and concerns
Reviewer 1
On behalf of all of the contributing authors, I would like to thank you for your constructive comments and for giving us the opportunity to revise our manuscript.
We sincerely thank you for the valuable feedback, which was very important to us, and we have carefully taken these suggestions into consideration in preparing our revisions and for improving the quality of our manuscript.
- Ascertainment of cases. Since data are self-reported, it is not clear to me how data on values of fasting glucose or HbA1c concentrations were gathered.
We apologize that this description was not clear in the original manuscript. All potential cases were contacted and asked to answer a diabetes-specific questionnaire that included questions on the circumstances of diagnosis (year of diagnosis, symptoms, biological exams, fasting, or random glucose concentrations at diagnosis), actual diabetes therapy (prescription of diet or physical activity, list of glucose lowering drugs taken), last measures of fasting glucose and Hb1Ac levels. We have now detailed the ascertainment of cases in the revised manuscript.
- If I understood well, there have been two different methods for defining incident of type 2 diabetes. Is this methodologically sound?
Thank you for this thoughtful comment. We have run a non-published sensitivity analysis excluding cases identified exclusively from self-report and we obtained similar results: indeed most cases identified by self-report before 2004 were later confirmed by medico-administrative data. Overall the main source of information is not the self-report but the medico-administrative database. The validation algorithm used in the E3N cohort to assess T2D cases has been largely accepted and used in several previous publications (e.g. Fagherazzi et al. Eur J Epidemiol. 2014 Nov;29(11):831-9; Mancini et al. Diabetologia. 2018 Feb;61(2):308-316; Mancini et al. Int J Hyg Environ Health. 2018 Aug;221(7):1054-1060.)
- I am not sure about the selection of dietary covariates. Authors controlled analyses for alcohol and coffee but they do not explain the reason. Moreover, I suggest the authors use a more global measurement of the diet quality, as for example a healthy dietary score.
Thank you for your suggestion. The selection of the covariates has been made a priori. We have reported this information in the method (statistical analysis) section. Page 3, line 131.
We have now specified in the method (covariables) section how the covariates were selected. “All potential covariates were selected a priori because of their known or suspected association with diabetes status and/or polyphenol intakes”
We also adjusted our main analysis on the ‘prudent’ diet and the result did not change subsequently.
- I cannot see any sub-group analysis. It would be interesting to test the robustness of main findings.
Thank you for leveraging this point. Our results are based on Bayesian profile regression. This method utilizes a Bayesian mixture model framework that takes into account the uncertainty associated with cluster assignments. The model is fitted using Markov chain Monte Carlo (MCMC) sampling methods and outputs a different clustering of the data at each iteration of the sampler. This method allows examining the “best” clustering of the data obtained from the algorithm and use model-averaging techniques to assess, using the posterior output obtained from the sampler and the uncertainty associated with subgroups contained within this “best” clustering. Finally, for the interpretation of our results, we rely on the credible interval of each cluster that was compared to the mean average of all clusters. Thus, we think that our main results were robust and their interpretations were done in a conservative way.
- The limitations of the study should be implemented also by clearly stating the limitation of self-reported dietary data.
We agree with this point and have changed the study limitation section. “In addition, our analyses are based on self-reported dietary consumption. In order to minimize the potential measurement error in the usual diet, we used a validated tool (13) and women with implausible diets were excluded.”
Reviewer 2 Report
This is an interesting and well-written manuscript. Minor corrections as follows are suggested:
Page 4, line 62 – “where” should be “were”
Page 4, line 79 – add a colon after “following”
Page 4, line 81 – the diagnostic criterion for diabetes in mmol/l A1c appears to be incorrect
Author Response
Response Letter
Dear Dr. Yoshimi Kishimoto,
Thank you for giving us the opportunity to submit a revised draft of our manuscript
entitled "Profiles of polyphenol intake and type 2 diabetes risk in 60,586 women followed for 20 years: results from the E3N cohort study" to nutrients. We appreciate the time and effort that you and the reviewers have dedicated to providing valuable feedback on this manuscript, and we are grateful to the reviewers for their insightful comments.
This revised manuscript incorporated changes to reflect the suggestions provided by the
reviewers. In this revised version, any revisions have been clearly highlighted using the
“Track Changes” function in Microsoft Word.
Here are our point-by-point responses to the reviewers’ comments and concerns
Reviewer 2
On behalf of all of the contributing authors, I would like to thank you for your constructive comments and for giving us the opportunity to revise our manuscript. These comments are all valuable and helpful for improving our article.
Accordingly, we have made modifications to our manuscript.
Page 4, line 62 – “where” should be “were”
Thank you for pointing out this mistake. We have corrected it.
Page 4, line 79 – add a colon after “following”
We apologize for this mistake, and the suggested correction has been made.
Page 4, line 81 – the diagnostic criterion for diabetes in mmol/l A1c appears to be incorrect
We apologize for this negligence. We have corrected the values. “In order to be considered as validated for type 2 diabetes, an individual must have reported at least one of the following: (1) fasting plasma glucose ≥ 7.0 mmol/l or random glucose ≥ 11.1 mmol/l at diagnosis; (2) use of a glucose-lowering medication; (3) most recent values of fasting glucose concentrations ≥ 7.0 mmol/l or HbA1c level ≥ 53 mmol/mol (7.0%) in the diabetes-specific questionnaire”